# In Silico Transcriptomic Expression of MSR1 in Solid Tumors Is Associated with Responses to Anti-PD1 and Anti-CTLA4 Therapies

**DOI:** 10.3390/ijms25073987

**Published:** 2024-04-03

**Authors:** Adrián Sanvicente, Cristina Díaz-Tejeiro, Cristina Nieto-Jiménez, Lucia Paniagua-Herranz, Igor López Cade, Győrffy Balázs, Víctor Moreno, Pedro Pérez-Segura, Emiliano Calvo, Alberto Ocaña

**Affiliations:** 1Experimental Therapeutics in Cancer Unit, Medical Oncology Department, Hospital Clínico San Carlos (HCSC), Instituto de Investigación Sanitaria (IdISSC), 28040 Madrid, Spain; adrian.sanvicente@salud.madrid.org (A.S.); cristina.diaztejeiro@salud.madrid.org (C.D.-T.); cnjietoj@salud.madrid.org (C.N.-J.); lucia.paniagua@salud.madrid.org (L.P.-H.); ilopez.7@alumni.unav.es (I.L.C.); 2Facultad de Ciencias Químicas, Universidad Complutense de Madrid, 28040 Madrid, Spain; 3Molecular Oncology Laboratory, Instituto de Investigación Sanitaria San Carlos (IdISSC), 28040 Madrid, Spain; 4Department of Bioinformatics, Semmelweis University, Tűzoltó u. 7-9, H-1094 Budapest, Hungary; gyorffy.balazs@yahoo.com; 5Research Centre for Natural Sciences, Hungarian Research Network, Magyar Tudosok Korutja 2, H-1117 Budapest, Hungary; 6START Madrid-Fundación Jiménez Díaz (FJD) Early Phase Program, Fundación Jiménez Díaz Hospital, 28040 Madrid, Spain; victor.moreno@startmadrid.com (V.M.); emiliano.calvo@startmadrid.com (E.C.); 7Medical Oncology Department, Hospital Clínico San Carlos (HCSC), Instituto de Investigación Sanitaria San Carlos (IdISSC), 28040 Madrid, Spain; pedro.perez@salud.madrid.org; 8START Madrid-HM Centro Integral Oncológico Clara Campal (CIOCC), Early Phase Program, HM Sanchinarro University Hospital, 28050 Madrid, Spain; 9Centro de Investigación Biomédica en Red en Oncología (CIBERONC), 28029 Madrid, Spain

**Keywords:** MSR1, anti-PD1, anti-CTLA4, solid tumors

## Abstract

Immuno-oncology has gained momentum with the approval of antibodies with clinical activities in different indications. Unfortunately, for anti-PD (L)1 agents in monotherapy, only half of the treated population achieves a clinical response. For other agents, such as anti-CTLA4 antibodies, no biomarkers exist, and tolerability can limit administration. In this study, using publicly available genomic datasets, we evaluated the expression of the macrophage scavenger receptor-A (SR-A) (MSR1) and its association with a response to check-point inhibitors (CPI). MSR1 was associated with the presence of macrophages, dendritic cells (DCs) and neutrophils in most of the studied indications. The presence of MSR1 was associated with macrophages with a pro-tumoral phenotype and correlated with TIM3 expression. MSR1 predicted favorable overall survival in patients treated with anti-PD1 (HR: 0.56, FDR: 1%, *p* = 2.6 × 10^−5^), anti PD-L1 (HR: 0.66, FDR: 20%, *p* = 0.00098) and anti-CTLA4 (HR: 0.37, FDR: 1%, *p* = 4.8 × 10^−5^). When specifically studying skin cutaneous melanoma (SKCM), we observed similar effects for anti-PD1 (HR: 0.65, FDR: 50%, *p* = 0.0072) and anti-CTLA4 (HR: 0.35, FDR: 1%, *p* = 4.1 × 10^−5^). In a different dataset of SKCM patients, the expression of MSR1 predicted a clinical response to anti-CTLA4 (AUC: 0.61, *p* = 2.9 × 10^−2^). Here, we describe the expression of MSR1 in some solid tumors and its association with innate cells and M2 phenotype macrophages. Of note, the presence of MSR1 predicted a response to CPI and, particularly, anti-CTLA4 therapies in different cohorts of patients. Future studies should prospectively explore the association of MSR1 expression and the response to anti-CTLA4 strategies in solid tumors.

## 1. Introduction

Immuno-oncology has experienced an increase in relevance with the approval of antibodies that have demonstrated clinical activity in different indications [1]. Among these compounds, we can include agents that target programmed death-ligand 1 (PD (L)1) or cytotoxic T-lymphocyte-associated protein 4 (CTLA4) molecules [2]. Very recently, an anti-lymphocyte-activation gene 3 (LAG3) antibody has also been approved for the treatment of patients with metastatic melanoma in the first-line setting [3]. These immunomodulatory proteins induce a suppressor action on effector T cells, therefore repressing a proper immune response [4]. In this context, patients who have greater responses to check-point inhibitors (CPI), mainly anti PD (L)1 agents, are those that have high expression levels of the PD1 ligand (PD L1) within their tumors [1]. Unfortunately, even using this selection criteria, a high number of patients do not respond to these therapies or they experience early progressions within the first months of treatment [5,6]. It is considered that even in a best-case scenario, the responses to CPIs alone will not benefit more than half of the treated population [5,6].

In parallel with this, no biomarker currently exists to predict the efficacy of anti-CTLA4 therapies. Therefore, the identification of molecular markers that can predict responses to these therapies is mandatory to better select patients that would benefit from these agents while avoiding undesirable side effects. Of note, it is very well known that this family of agents has poorer tolerability than anti-PD (L)1 compounds.

Some tumors display an immunosuppressive environment where the action of immune effector cells is repressed through the presence of different CPIs [2]. In addition to the negative effect of tumoral cells on the activation of effector cells, some immune cells such as myeloid-derived cells or Tregs also play a role promoting an immunosuppressive environment [7,8]. In line with this, some cells that belong to innate immunity have demonstrated the potential to play a negative role, as confirmed when agents that act on some of their suppressor receptors augment the efficacy of anti-*PD (L) 1* agents [9]. An additional example is the presence of M2 or pro-tumoral macrophages that prevent an effector immune response and have no antigen-presenting capacity [10,11,12].

*MSR1*, also known as macrophage scavenger receptor-A (SR-A) or cluster of differentiation 204 (CD204), is mainly associated with the uptake and degradation of acetylated low-density lipoprotein (acetyl-LDL) but not non-modified low-density lipoprotein (LDL) [13,14,15]. This produces an increase in intracellular cholesterol deposition, leading to effects similar to those observed in familial hypercholesterolemia [16,17]. Beyond this role, *MSR1* has been implicated in different human pathologies and very recently in cancer progression [18,19]. The expression of *MSR1* has been linked to significantly poor prognoses and the increased severity of multiple forms of cancer [19,20].

In this article, we evaluated the expression of *MSR1* in cancer, observing an association with the presence of macrophages, DCs and neutrophils, in addition to a clear prediction of the response to anti-PD (L) 1 and CTLA4 inhibitors.

## 2. Results

### 2.1. MSR1 Is Highly Expressed in Solid Tumors

We first mapped the expression of *MSR1* on all available tumor types using publicly available datasets, as described in the material and methods section and in Appendix A. *MSR1* was highly expressed, with a statistically significant difference in some tumors, including glioblastoma (GBM), kidney renal cell carcinoma (KIRC), breast cancer (BRCA), pancreatic adenocarcinoma (PAAD), ovarian carcinoma (OV), stomach adenocarcinoma (STAD), skin cutaneous melanoma (SKCM) and esophagus carcinoma (ESCA), among others (Figure 1A,B). On the other hand, we observed a higher expression in normal tissue compared with transformed tissue in lung adenocarcinoma (LUAD) and lung squamous cell carcinoma (LUSC) (Figure 1A). The tumors with the highest fold changes compared with normal tissues included GBM (≥100), followed by low-grade glioma (LGG) and PAAD, which had ≥10-fold changes. ESCA, STAD, SKCM and OV displayed ≥5-fold changes, and KIRCC, diffuse large B cell lymphoma (DLBCL), thymoma (THYM), head and neck squamous cell carcinoma (HNSCC) and BRCA had ≥2-fold changes (Figure 1B). We did not observe relevant differences between the breast cancer subtypes or ovarian cancer subgroups (Appendix A). For further analysis, we set exclusion criteria based on a statistically significant difference between the tumor and normal tissue and a fold change expression equal to or greater than two. With these criteria, we selected the following tumor types for further study: GBM, PAAD, ESCA, STAD, SKCM, OV, KIRC, THYM, HNSC and BRCA. We excluded DLBC as we intended to focus only on solid tumors.

### 2.2. MSR1 Expression Is Correlated with The Presence of Innate Immune Cells

In a second step, we aimed to explore the expression of *MSR1* in relation to the presence of several immunologic cells. We evaluated the following ten selected tumor types: GBM, PAAD, ESCA, STAD, SKCM, OV, KIRC, THYM, HNSC and BRCA, following the criteria described before. As can be seen in Figure 2, a clear and very strong association was observed between the expression levels of *MSR1* and the presence of macrophages in almost every tumor type studied, with SKCM being the one with a weaker but still positive correlation (GBM: Rho = 0.645, *p* = 1.82× 10^17^; PAAD: Rho = 0.640, *p* = 4.28 × 10^−21^; ESCA: Rho = 0.617, *p* = 3.14 × 10^−20^; STAD: Rho = 0.675, *p* = 1.13 × 10^−51^; SKCM: Rho = 0.289, *p* = 2.94 × 10^−10^; OV: Rho = 0.81, *p* = 3.30 × 10^−59^; KIRC: Rho = 0.536, *p* = 1.30 × 10^−35^; THYM: Rho = 0.493, *p* = 2.14 × 10^−8^; HNSC: Rho = 0.513, *p* = 2.40 × 10^−34^; and BRCA: Rho = 0.579, *p* = 3.68 × 10^−90^). A similar association was identified for the DCs in all tumor types, except for THYM (GBM: Rho = 0.519, *p* = 7.89 × 10^−11^; PAAD: Rho = 0.791, *p* = 6.86 × 10^−38^; ESCA: Rho = 0.578, *p* = 1.93 × 10^−17^; STAD: Rho = 0.708, *p* = 5.71 × 10^−59^; SKCM: Rho = 0.369, *p* = 3.50 × 10^−16^; OV: Rho = 0.502, *p* = 2.47 × 10^−17^; KIRC: Rho = 0.610, *p* = 2.44 × 10^−48^; THYM: Rho = −0.320, *p* = 4.81 × 10^−4^; HNSC: Rho = 0.360, *p* = 1.53 × 10^−16^; and BRCA: Rho = 0.282, *p* = 1.21 × 10^−19^) and in the neutrophils (GBM: Rho = 0.609, *p* = 2.80 × 10^−15^; PAAD: Rho = 0.694, *p* = 6.39 × 10^−26^; ESCA: Rho = 0.455, *p* = 1.39 × 10^−10^; STAD: Rho = 0.477, *p* = 6.30 × 10^−23^; SKCM: Rho = 0.646, *p* = 3.08 × 10^−55^; OV: Rho = 0.557, *p* = 1.17 × 10^−21^; KIRC: Rho = 0.738, *p* = 2.22 × 10^−80^; THYM: Rho = −0.081, *p* = 3.90 × 10^−1^; HNSC: Rho = 0.581, *p* = 9.22 × 10^−46^; and BRCA: Rho = 0.525, *p* = 1.75 × 10^−71^) (Figure 2A,B).

A strong association with Tregs was only observed for PAAD (Rho = 0.613), and mild associations were observed for STAD (Rho = 0.443), HNSC (Rho = 0.527) and BRCA (Rho = 0.425). There were also associations with CD8 T cells in some tumor types, such as PAAD (Rho = 0.604) and STAD (Rho = 0.669) (Figure 2A). Weaker correlations were identified for the other immune cells evaluated, including B cells and T CD4.

Of note in all these settings, low purity scores were detected, and this was what demonstrated that *MSR1* was mainly present in immune cells and not in tumoral cells (Appendix A).

### 2.3. Correlation of MSR1 Expression Level with Macrophage Subtypes and Immune-Suppressive Molecules

As MSR1 has been described as a classical receptor expressed in macrophages, we explored whether its presence was particularly associated with a specific macrophage subtype. As can be seen in Figure 3A, a clear correlation was observed with the expression of *MSR1* and the presence of M2 macrophages in the following particular tumor types [10,11,12]: GBM (Rho = 0.427), PAAD (Rho = 0.801), ESCA (Rho = 0.759), STAD (Rho = 0.827), SKCM (Rho = 0.684), OV (Rho = 0.770), KIRC (Rho = 0.746), THYM (Rho = 0.451), HNSC (Rho = 0.781) and BRCA (Rho = 0.717). Tumor-associated macrophages are typically classified as M2s, and they usually do not present antigens to T cells promoting the presence of immune-suppressive molecules. With the intent of characterizing the immune microenvironment associated with the presence of *MSR1*, we explored the association of *MSR1* with different molecules related to immune activation or suppression. As can be seen in Figure 3B, a strong association was observed for the presence of *TIM3* with GBM (Rho = 0.727), PAAD (Rho = 0.906), ESCA (Rho = 0.824), STAD (Rho = 0.879), SKCM (Rho = 0.813), OV (Rho = 0.893), THYM (Rho = 0.643), HNSC (Rho = 0.832) and BRCA (Rho = 0.857). Of note, there were also positive correlations for 4-1BB with GBM (Rho = 0.424), PAAD (Rho = 0.643), KIRC (Rho = 0.529), THYM (Rho = 0.492) and BRCA (Rho= 0.512), and for PDL1, there were positive correlations with PAAD (Rho = 0.607), STAD (Rho = 0.484), SKCM (Rho = 0.544), OV (Rho = 0.443), KIRC (Rho = 0.406) and BRCA (Rho = 0.489). Other immune activators, expressed in B cells and DC such as *OX40* and *CD40,* did not correlate with the presence of *MSR1* (Figure 3B).

Finally, in the available tumor types including PAAD, SKCM and BRCA, we evaluated the co-expression of *MSR1* and co-stimulatory molecules for the same cell types in macrophages and DCs using single cell analysis (Figure 3C). Of note, no co-expression was identified in those cells, demonstrating that *TIM3*, *4-1BB* and *PDL1* were not expressed in the same cell lines. These data globally describe the presence of *MSR1* in pro-tumoral or M2 macrophages and its association with co-inhibitory molecules such as *TIM3*, *4-1BB* and *PDL1*.

### 2.4. MSR1 Predicts Favorable Outcomes in Patients Treated with Check-Point Inhibitors

Next, we assessed the role of *MSR1* in the response to CPIs. To do so, we used publicly available data from patients treated with anti-PD(L) 1 and anti-CTLA4 antibodies. As can be seen in Figure 4A, the patients treated with CPIs, including anti-PD(L)1 or CTLA4, had favorable survival rates, as follows: anti-PD1 (HR: 0.56, FDR: 1%, *p* = 2.6 × 10^−5^), anti-PD-L1 (HR: 0.66, FDR:20%, *p* = 0.00098) and anti-CTLA4 (HR: 0.37, FDR:1%, *p* = 4.8 × 10^−5^). Of note, 55% of the samples included in this analysis belonged to the same tumor types evaluated earlier. We then aimed to confirm the data in a specific tumor type, where a higher number of patients were available. As can be seen in Figure 4B, the expression of *MSR1* was associated with favorable outcomes in the SKCM patients treated with CPIs, as follows: anti-PD1 (HR: 0.65, FDR: 50%, *p* = 0.0072) and anti-CTLA4 (HR: 0.35, FDR: 1%, *p* = 4.1 × 10^−5^). Finally, we compared MSR1 expression with PD-L1 expression as a potential biomarker, using the same population. As shown in Figure 4C, we could observe that PD-L1 was associated with the response to CPIs with a similar accuracy as MSR1 was for anti-PD (L)1 treatment, and much poorer response was associated with anti-CTLA4, as follows: anti-PD1 (HR: 0.54, FDR: 1%, *p* = 6.3 × 10^−5^), anti-PD-L1 (HR: 0.54, FDR: 1%, *p* = 2.8 × 10^−7^) and anti-CTLA4 (HR: 0.51, FDR: 50%, *p* = 0.0078).

### 2.5. MSR1 Expression Predicts Clinical Response in a Different Dataset

To confirm the previous findings, we used a different dataset to evaluate whether patients treated with CPIs experienced a better clinical response based on their expression levels of *MSR1*. As can be seen in Figure 5A, high expression levels of *MSR1* predicted the response to CPI: for the response to any CPI we found: AUC: 0.552, *p* = 2.5 × 10^−3^; for the response to anti-PD1, we found AUC: 0.581, *p* = 1.3 × 10^−3^; and for anti-CTLA4, we found AUC: 0.612, *p* = 2 × 10^−2^. In a specific cohort of SKCM patients, we observed results that trended in the same direction, as follows: for any CPI therapy, AUC: 0.56, *p* = 2.6 × 10^−2^; for anti-PD1, AUC: 0.556, *p* = 4.9 × 10^−2^; and for anti-CTLA4, AUC: 0.61, *p* = 2.9 × 10^−2^ (Figure 5B).

## 3. Discussion

In the current study, we mapped the presence of *MSR1* in solid tumors. The tumor types with the highest MSR1 expression levels included GBM, KIRC, BRCA, PAAD, OV, STAD, SKCM and ESCA, among others. The tumors with the highest fold-change differences compared with normal tissue included GBM, LGG and PAAD. Of note, the expression level of *MSR1* in non-transformed lung tissue was higher than that in the tumors.

When evaluating cells related to the expression of *MSR1,* we observed that a negative correlation with tumor purity was observed, suggesting that the protein was mainly present in the tumor’s microenvironment. Indeed, positive associations were identified with macrophages, DCs and neutrophils. No associations with adaptive cells, including T or B cells, were observed. These data indicated that *MSR1* mRNA was mainly present in these cells in the tumors’ microenvironments, except for lung cancer, where it existed at high levels in immune cells within the normal tissue. These findings are in line with the literature regarding several normal and physiological conditions beyond cancer, and they reinforce the concept and role of antigen-presenting cells (APCs) in non-transformed lung [21].

We next observed an association between MSR1 expression and M2 or pro-tumoral macrophages [22,23]. This was in line with previous publications that reported the presence of MSR1 in tumor-associated macrophages (TAM) and correlated that presence with detrimental prognoses for several tumor types [22,23,24,25,26,27,28,29]. Other studies have described their expression with high-grade metastases and tumor aggressiveness [19].

Positive correlations with *TIM3* and *PD-*L1 were identified in the tumor types studied, signifying that MSR1 is abundant in an immune-suppressive environment. To our knowledge, only one study on glioma has suggested the co-expression of *MSR1* and *TIM3* [28]. A single-cell analysis showed that TIM3 was not present in the same cell populations with higher levels of *MSR1* transcripts, with mainly macrophages or dendritic cells. Studies describing an association between *PDL1* expression and *MSR1* have been published, confirming our findings [30]. An interesting observation was the presence of *4-1BB*, which could suggest the existence of already activated and exhausted T cells.

Our study discovered the association of *MSR1* expression with response to CPIs, particularly anti-CTLA4 antibodies. Indeed, the presence of MSR1 predicted outcome with more accuracy than PD-L1, particularly for anti-CTLA4 therapies, opening the possibility for this biomarker to be specifically evaluated in patients treated with anti-CTLA4 therapies.

Finally, we used melanoma as a model tumor type to confirm the capacity to predict the response to CPI and, particularly, anti-CTLA4 antibodies. In this context, given the fact that there are no available biomarkers for predicting the response to anti-CTLA4 therapies, the identification of markers that could help to select patients in this clinical setting is a priority, particularly given the toxicity observed with anti-CTLA4 agents.

We acknowledge that our study has limitations. This was an analysis performed using published available datasets in which bulk and single-cell information was extracted. In this context, the confirmation of these data in human samples using immunohistochemistry techniques would undoubtedly help to confirm these results. In addition, we recognize that for the outcome analysis, we could not evaluate some of the tumor types that we used for the correlational study. However, the analysis performed included sixty percent of the populations with those tumor types, and we also used melanoma as a tumor type to confirm the data identified in the whole population.

In conclusion, we described as a potential novel biomarker associated with the response to CPIs, particularly anti-CTLA4, that is mainly expressed in antigen-presenting cells (M2 macrophages) and that is present in an immunosuppressive microenvironment. Further studies should be performed to confirm MSR1’s prediction capacity in patients.

## 4. Materials and Methods

### 4.1. Data Collection and Processing

Processed TCGA (The Cancer Genome Atlas; https://www.cancer.gov/ccg/research/genome-sequencing/tcga; last accessed 23 January 2023) PanCancer data were downloaded using different web tools. This dataset contains whole exome sequencing and RNA-Seq information from patients’ tumors and their matched normal tissues. The publicly available web tools that were used to evaluate the expression of the selected genes and the mutational information included Gepia2 [31] (http://gepia2.cancer-pku.cn/#index; last accessed: 23 January 2023), Gent2 [32] (http://gent2.appex.kr/gent2/; last accessed 23 January 2023) and CBioportal [33] (https://cbioportal.org, last accessed 24 January 2023). Tumor types where a fold change greater than 2 between the expression of MSR1 in the tumor vs. normal tissues were selected.

### 4.2. Immune Cell Infiltration and Gene Expression Correlation

The Tumor Immune Estimation Resource (TIMER) platform [23] was used to investigate the association between gene expression and tumor purity and the association between the presence of tumor immune infiltrates, including CD4+ T cells, CD8+ T cells, Tregs, macrophages, dendritic cells, neutrophils and B cells. TIMER applies a deconvolution method (previously described in [34,35] (http:/timer.cistrome.org/, accessed 25 January 2023)) to infer the abundance of tumor-infiltrating immune cells from gene expression profiles. It contains 10,897 samples from diverse cancer types from the TCGA.

Single-cell data were obtained from the MNP-VERSE Seurat package (downloaded from https://gustaveroussy.github.io/FG-Lab/ accessed 17 February 2023) [36]. The data were preprocessed by filtering and normalizing using the Seurat R package. Cell clustering, dimensionality reduction and visualization were performed using R software (version 4.1.0).

### 4.3. Gene Correlations

The Spearman correlation coefficients between the expressions of every pair of genes were used for the correlation analysis between them. Data from TCGA [37] were included in the analysis.

### 4.4. Outcome and Prognosis Analysis

For the evaluation of the activities of the CPIs, we used datasets that included patients treated with these agents. The datasets were identified in the gene expression omnibus (GEO) using the keywords “gene expression”, “PD1”, “CTLA4” and “immunotherapy”, as well as the names of the available immunotherapy agents, as described elsewhere by members of this research group [38]. In this cohort, we evaluated the association of *MSR1* expression with overall survival (OS). Patients were separated into two cohorts according to the best cut-off values and analyzed based on the administered therapy, and the anti-PD1 treatment population included 797 patients while the anti-CTLA4 cohort included 131 patients. The Kaplan–Meier (KM) plots were presented with hazard ratios (HRs), 95% confidence intervals (CIs) and log-rank *p*-values (*p*). Genes with an HR of <1 and a *p*-value of <0.05 were considered predictors of favorable outcomes, while genes with an HR of >1 and a *p*-value of <0.05 were considered predictors of detrimental outcomes. This was performed with the KM Plotter Online tool [39,40] (https:/kmplot.com/analysis/, accessed 25 January 2023). As described in the KM Plotter to avoid missing correlations due to the use of a specific cut-off, all available cut-off values between the lower and upper quartiles of expression were used for the selected genes (as binary variables), and false discovery rate (FDR) using the Benjamini–Hochberg method was computed to correct for the multiple hypothesis testing. The cut-off value with the highest significance (the lowest FDR value) was determined. The ROC plotter online tool [41] was used to correlate gene expression and response with immunotherapy (anti-*PD1* or anti-*CTLA4*) in a cohort of different solid tumors that included metastatic and primary tumors. The area under the curve (AUC) was computed to evaluate the clinical utility (the sensitivity and specificity) of MSR1 as a biomarker. An AUC of 0.5 corresponded to no classification power at all, while an AUC value of 1 denoted a perfect biomarker.

### 4.5. Datasets Used

Information describing all datasets used in this analysis is provided in Appendix A.

### 4.6. Graphical Design

The histograms, bar charts and heatmaps were plotted using GraphPad Prism software (Prism 9.5.1—7 February 2023) (GraphPad Software, San Diego, CA, USA).

## Figures and Tables

**Figure 1 ijms-25-03987-f001:**
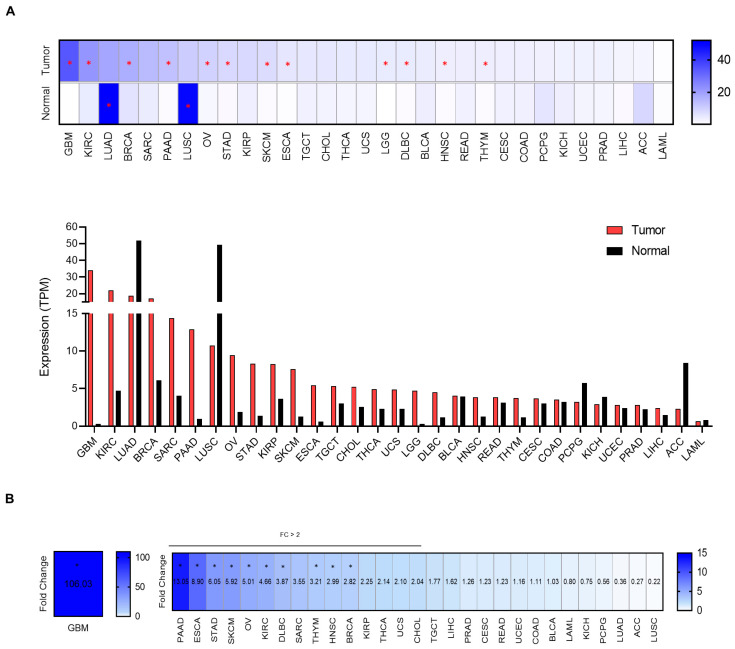
*MSR1* expression profiles across all tumor samples and paired normal tissues. (**A**) *MSR1* expression as transcript per million (TPM) in several tumor types represented using a heatmap and bar graph using the GEPIA2 database. Statistically significant differences between the normal tissue and tumoral ones are marked by an asterisk. (**B**) Heatmap showing the fold changes (FCs) between the expression of *MSR1* in the tumoral and normal tissues for the previous tumor types. The inclusion criteria for the selection of cancers for further analysis were set as a FC of >2 and a statistical difference between the expression in the tumor and normal tissues.

**Figure 2 ijms-25-03987-f002:**
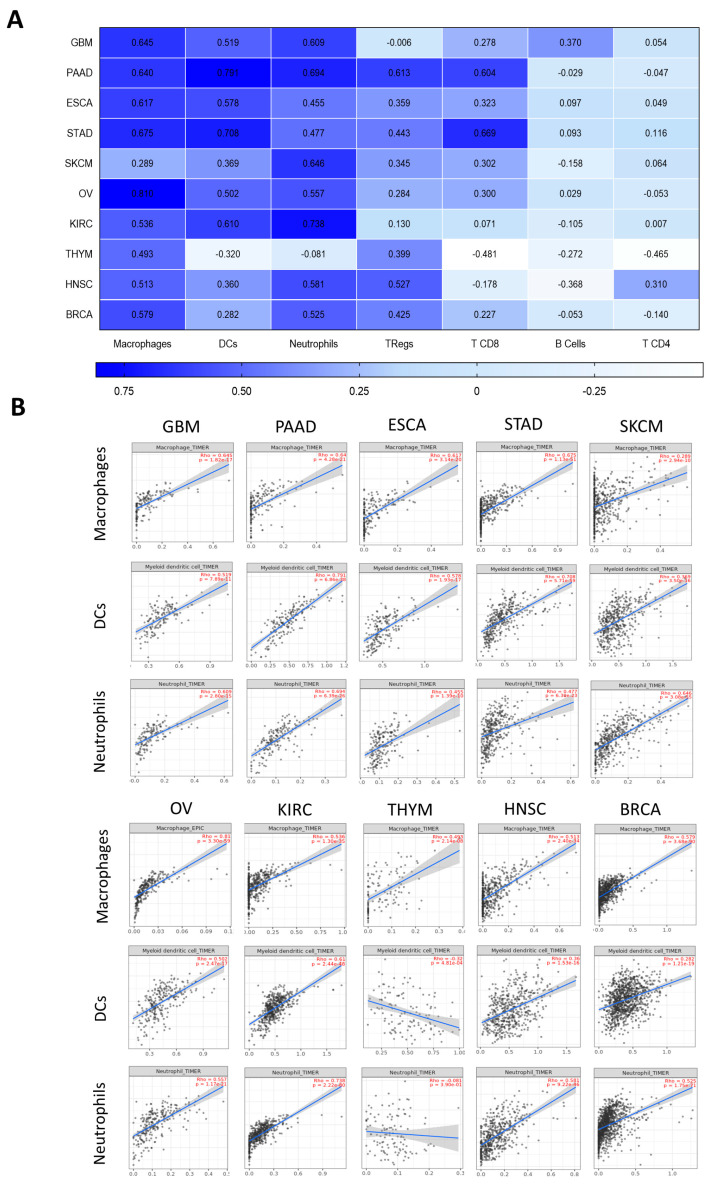
Association of *MSR1* expression levels with immune infiltrates in the ten selected cancer types. (**A**) Correlation analysis between *MSR1* expression and relevant immune population infiltrates in different tumor types using TIMER2.0. A Spearman’s correlation was used with a purity adjustment. (**B**) Dot plot details of the correlations displayed in (**A**): macrophages, DCs and neutrophils. The *MSR1* expression level is represented on the Y axis as log2 (transcripts per million, TPM) and the population infiltration level is shown on the X axis.

**Figure 3 ijms-25-03987-f003:**
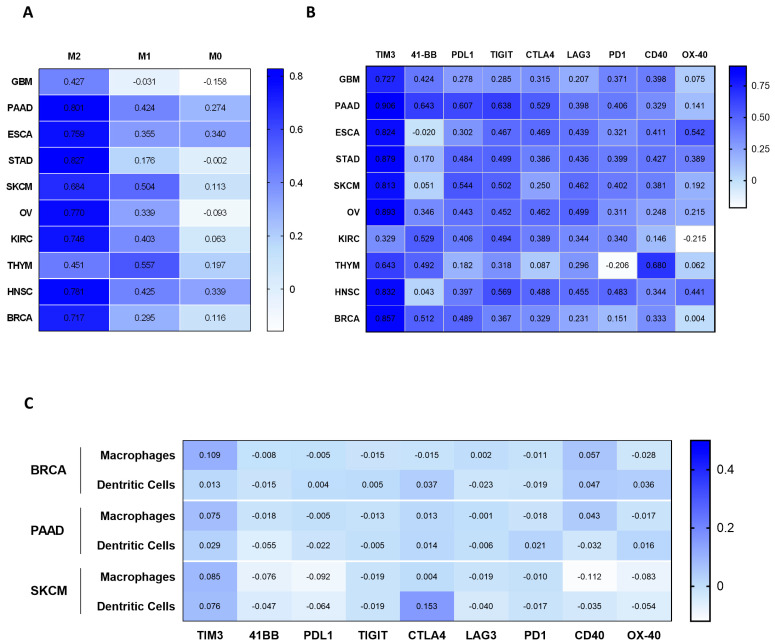
Association of *MSR1* expression levels with macrophage subtypes (M0, M1 and M2 classifications) and co-stimulatory immune checkpoints for the ten selected cancer types. (**A**) Correlation analysis between *MSR1* expression and macrophage subtypes (M2, M1 and M0). (**B**) Correlation between the expression of *MSR1* and immunomodulatory proteins. (**C**) Single cell correlation analysis between *MSR1* and the co-inhibitory genes described in (**B**) for those tumors with available data.

**Figure 4 ijms-25-03987-f004:**
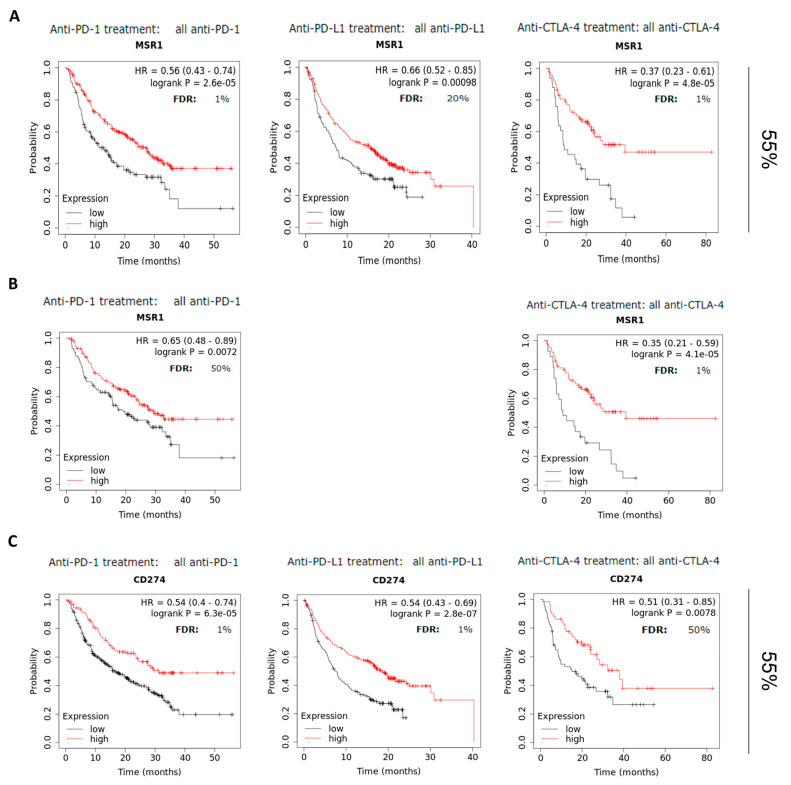
Association of *MSR1* expression levels with overall survival (OS) in patients treated with immune check-point inhibitors (ICI), including anti-PD1, anti-PDL1 and anti-CTLA4, respectively. (**A**) Kaplan–Meier survival plots comparing the high vs. low expression levels of *MSR1,* including the patients with all available tumors. (**B**) Kaplan–Meier survival plots comparing the high vs. low expression levels of *MSR1* for the SKCM patients. (**C**) Kaplan–Meier survival plots comparing the high vs. low expression levels of CD274 with OS, including the patients with all available tumors.

**Figure 5 ijms-25-03987-f005:**
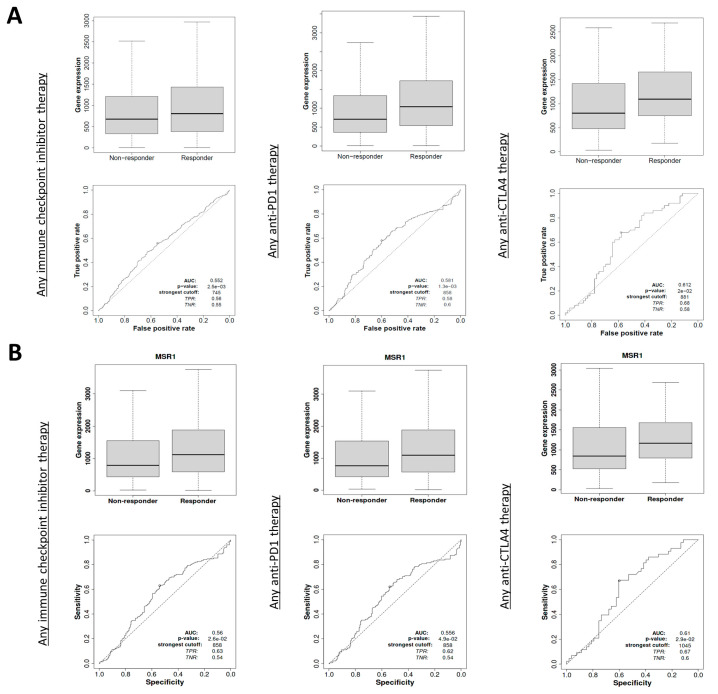
Association of *MSR1* expression levels with responses in patients treated with ICIs including anti-PD1, anti-PDL1 and anti-CTLA4 (measured using area under the curve (AUC) and *p* values). The data were obtained using ROCPLOT. (**A**) For all available cancers. (**B**) For the SKCM patients.

## Data Availability

All data generated or analyzed during this study are included in this published article [and its Appendix A].

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
