# Peer review of "In Silico Transcriptomic Expression of MSR1 in Solid Tumors Is Associated with Responses to Anti-PD1 and Anti-CTLA4 Therapies"

_ijms, 2024, doi:10.3390/ijms25073987_

Round 1

Reviewer 1 Report

Comments and Suggestions for Authors

Authors describe an association of MRS1 expression and particular populations of immune cells and immune suppressive molecules.  They propose that expression of MRS1 can also show correlations with outcome in patients treated with check point inhibitors. These proposals are based solely on statistical analyses of data from publicly available genomic datasets and no experimental validation were made (for example, with western blot or immunohistochemistry). The manuscript represents a start point for the study of MRS1 expression as a potential biomarker.

The study has some merit because of the theoretical predictions, however no experimental validations were made.

In my opinion, some minor changes must be made before the manuscript can be accepted for publication.

1.        The figure legends are incomplete. A figure must be self-explanatory and Figure legends must contain the meaning of all abbreviations and indicate what the numbers in cells represent. The Figure legend of Figure 2A must indicate what TPM is.

2.        Along the text authors use the word “gene” as a tool for their associations. However, this is not accurate. A gen can be present in a cell or tissue but can or cannot be expressed. In the correlations, authors should use something like “expression levels” or “mRNA levels” or “protein levels”. The use of the word “gene” in the manuscript is really confusing, it can be interpreted as copy number of segments of DNA, but not the product of a gene (mRNA or protein). Please rephrase sentences and change the word “gene” for something more accurate (“mRNA” or “protein levels”). For example, in line 210 the sentence “this gene was mainly present in the tumor microenvironment” This is really weird because in somatic cells, all the genes are present, some are expressed and others are silenced but all are present. It would be more accurate something like “this protein was mainly present in the tumor microenvironment”.

3.        Figure 4. It is not clear why authors sometimes use CD274 and sometimes PD-L1. The same name must be used in all the document.

4.        Line 249. The sentence “describe MSR1 as a novel biomarker” must be changed to “as a potential biomarker”.

5.        In the introduction section, the name and function of all the immunomodulatory PROTEINS (not genes) and immune cells mentioned must be described. For example TIGIT, LAG3 and others of Figure 3.

Author Response

Comments and Suggestions for Authors

Authors describe an association of MRS1 expression and particular populations of immune cells and immune suppressive molecules.  They propose that expression of MRS1 can also show correlations with outcome in patients treated with check point inhibitors. These proposals are based solely on statistical analyses of data from publicly available genomic datasets and no experimental validation were made (for example, with western blot or immunohistochemistry). The manuscript represents a start point for the study of MRS1 expression as a potential biomarker.

The study has some merit because of the theoretical predictions, however no experimental validations were made.

In my opinion, some minor changes must be made before the manuscript can be accepted for publication.

  1. The figure legends are incomplete. A figure must be self-explanatory and Figure legends must contain the meaning of all abbreviations and indicate what the numbers in cells represent. The Figure legend of Figure 2A must indicate what TPM is.

Response: we apologies if some information is missing in the figures. We have added all information required to make figures comprehensive and easy to follow.

  1. Along the text authors use the word “gene” as a tool for their associations. However, this is not accurate. A gen can be present in a cell or tissue but can or cannot be expressed. In the correlations, authors should use something like “expression levels” or “mRNA levels” or “protein levels”. The use of the word “gene” in the manuscript is really confusing, it can be interpreted as copy number of segments of DNA, but not the product of a gene (mRNA or protein). Please rephrase sentences and change the word “gene” for something more accurate (“mRNA” or “protein levels”). For example, in line 210 the sentence “this gene was mainly present in the tumor microenvironment” This is really weird because in somatic cells, all the genes are present, some are expressed and others are silenced but all are present. It would be more accurate something like “this protein was mainly present in the tumor microenvironment”.

Response: we completely agree with the reviewer; the term “gene” can be misleading. Therefore, we have modified and changed this term along the manuscript for other words that are more accurate like “transcript” or “mRNA”.

  1. Figure 4. It is not clear why authors sometimes use CD274 and sometimes PD-L1. The same name must be used in all the document.

Response: we apologies for this. We have added PD-L1 along the manuscript.

  1. Line 249. The sentence “describe MSR1 as a novel biomarker” must be changed to “as a potential biomarker”.

Response: This has been modified accordingly.

  1. In the introduction section, the name and function of all the immunomodulatory PROTEINS (not genes) and immune cells mentioned must be described. For example, TIGIT, LAG3 and others of Figure 3.

Response: this has been modified accordingly.

Introduction and methods can be improved.

Response: We have completed the material and methods section adding additional information

Reviewer 2 Report

Comments and Suggestions for Authors

This article provides a detailed analysis on the expression of macrophage scavenger-A (SR-A) receptor (MSR1) in different types of solid tumors and its association with response to immune checkpoint inhibitor (CPI) treatments. The authors used publicly available genomic data to evaluate MSR1 expression and its correlation with the presence of innate immune cells, macrophage subtype, and the expression of immune-suppressive molecules such as TIM3 and PD-L1.

To confirm without doubt the results obtained and investigate the validity of MSR1 as a predictive biomarker of response to CPI, it would be useful to conduct the following experiments:

1. Validation of results with immunohistochemistry on human samples: To confirm the expression of MSR1 and its association with response to CPIs, it would be essential to perform immunohistochemistry analyzes on human tumor tissue samples. This would allow us to confirm the results obtained using public genomic data.

2. Functional experiments to understand the role of MSR1 in tumor macrophages: It would be useful to conduct in vitro experiments to understand the role of MSR1 in regulating macrophage function in the tumor microenvironment. This could include genetic manipulations to modulate MSR1 expression and evaluate its impact on macrophage polarization and the production of immunosuppressive molecules.

3. Experiments in animal models: To better understand the role of MSR1 in the response to CPI treatments and evaluate its potential as a therapeutic target, it would be useful to conduct studies in animal models of solid tumors. These studies could include the use of xenograft models or spontaneous tumor transplantation models in MSR1 knockout mice.

In summary, while the article provides preliminary evidence for the association of MSR1 with response to CPIs, further experiments are needed to confirm and investigate these findings and evaluate the potential of MSR1 as a predictive biomarker and therapeutic target in solid tumors.

Author Response

Comments and Suggestions for Authors

This article provides a detailed analysis on the expression of macrophage scavenger-A (SR-A) receptor (MSR1) in different types of solid tumors and its association with response to immune checkpoint inhibitor (CPI) treatments. The authors used publicly available genomic data to evaluate MSR1 expression and its correlation with the presence of innate immune cells, macrophage subtype, and the expression of immune-suppressive molecules such as TIM3 and PD-L1.

To confirm without doubt the results obtained and investigate the validity of MSR1 as a predictive biomarker of response to CPI, it would be useful to conduct the following experiments:

  1. Validation of results with immunohistochemistry on human samples: To confirm the expression of MSR1 and its association with response to CPIs, it would be essential to perform immunohistochemistry analyzes on human tumor tissue samples. This would allow us to confirm the results obtained using public genomic data.

Response: we completely agree with the reviewer. We are about to start a project in our institution evaluating the presence of this protein in relation with response to CPIs. We have started to select the patient population. We hope that this data will be available during the next year. 

  1. Functional experiments to understand the role of MSR1 in tumor macrophages: It would be useful to conduct in vitro experiments to understand the role of MSR1 in regulating macrophage function in the tumor microenvironment. This could include genetic manipulations to modulate MSR1 expression and evaluate its impact on macrophage polarization and the production of immunosuppressive molecules.

Response: We don’t have any doubt that these experiments will add enormous information regarding our findings and will be the best manner to characterize the functional role of MSR1. Although we have planned these experiments and we are preparing the methods for the execution, we will not be able to have it done in less than 6-8 months, in the best case scenario.

  1. Experiments in animal models: To better understand the role of MSR1 in the response to CPI treatments and evaluate its potential as a therapeutic target, it would be useful to conduct studies in animal models of solid tumors. These studies could include the use of xenograft models or spontaneous tumor transplantation models in MSR1 knockout mice.

Response: In line with the previous response, we also agree with this suggestion. However, as mentioned, this kind of experiments can not be executed in less than 6-8 months.

In summary, while the article provides preliminary evidence for the association of MSR1 with response to CPIs, further experiments are needed to confirm and investigate these findings and evaluate the potential of MSR1 as a predictive biomarker and therapeutic target in solid tumors.

Reviewer 3 Report

Comments and Suggestions for Authors

The authors presented “Genomic expression of MSR1 in solid tumors associates with response to anti-PD1 and anti-CTLA4 therapies”. This study thoroughly examines MSR1 expression across various solid tumors, shedding light on its correlation with immune cells and phenotypes, advancing our understanding of the tumor microenvironment and its relevance for immunotherapy. Identifying MSR1 as a potential biomarker for predicting checkpoint inhibitor response, particularly in anti-CTLA4 therapies, fills a critical gap in immuno-oncology, promising personalized treatment and better patient outcomes. Leveraging publicly available genomic datasets enhances the study's credibility, enabling robust validation across diverse patient cohorts, while rigorous statistical analysis bolsters the reliability of the findings. The observed association between MSR1 expression and improved overall survival in patients undergoing anti-PD1, anti-PD-L1, and anti-CTLA4 therapies underscores the study's clinical significance, potentially guiding treatment decisions and refining patient selection for immunotherapy. Acknowledging the need for future prospective studies to explore MSR1's relationship with anti-CTLA4 strategies demonstrates the study's forward-looking perspective, highlighting promising avenues for further research and clinical exploration in immuno-oncology.

Author Response

Comments and Suggestions for Authors

The authors presented “Genomic expression of MSR1 in solid tumors associates with response to anti-PD1 and anti-CTLA4 therapies”. This study thoroughly examines MSR1 expression across various solid tumors, shedding light on its correlation with immune cells and phenotypes, advancing our understanding of the tumor microenvironment and its relevance for immunotherapy. Identifying MSR1 as a potential biomarker for predicting checkpoint inhibitor response, particularly in anti-CTLA4 therapies, fills a critical gap in immuno-oncology, promising personalized treatment and better patient outcomes. Leveraging publicly available genomic datasets enhances the study's credibility, enabling robust validation across diverse patient cohorts, while rigorous statistical analysis bolsters the reliability of the findings. The observed association between MSR1 expression and improved overall survival in patients undergoing anti-PD1, anti-PD-L1, and anti-CTLA4 therapies underscores the study's clinical significance, potentially guiding treatment decisions and refining patient selection for immunotherapy. Acknowledging the need for future prospective studies to explore MSR1's relationship with anti-CTLA4 strategies demonstrates the study's forward-looking perspective, highlighting promising avenues for further research and clinical exploration in immuno-oncology.

Response: We absolutely agree with the reviewer's comment on the need for future studies, which we intend to pursue soon, as well as the potential of this initial study to better understand the tumor microenvironment in these complex cases. We really appreciate your comments and have done our best to improve our work and look forward to continuing it.

Round 2

Reviewer 2 Report

Comments and Suggestions for Authors

It would be appropriate to change the title of the manuscript to better reflect the content and objective of the study. Considering the time required to conduct further studies, the manuscript can be accepted in its current form. However, to definitively confirm the results and evaluate the role of MSR1 as a predictive biomarker of response to CPIs, further experiments are needed. These include validation with immunohistochemistry on human samples, functional experiments to understand the role of MSR1 in tumor macrophages, and studies in animal models.
